# Efficacies of Nitrogen Removal and Comparisons of Microbial Communities in Full-Scale (Pre-Anoxic Systems) Municipal Water Resource Recovery Facilities at Low and High COD:TN Ratios

**Supaporn Phanwilai [1], Pongsak (Lek) Noophan [2,\*], Chi-Wang Li [3] and Kwang-Ho Choo [4]**

[1] Department of Knowledge of The Land for Sustainable, School of Integrated Science, Kasetsart University, Bangkok 10900, Thailand; supaporn.phan@ku.th

[2] Department of Environmental Engineering, Faculty of Engineering, Kasetsart University, Bangkok 10900, Thailand

[3] Department of Water Resources and Environmental Engineering, Tamkang University, New Taipei City 25137, Taiwan; chiwang@mail.tku.edu.tw

[4] Department of Environmental Engineering, Kyungpook National University, 80 Daehak-ro, Buk-gu, Daegu 702-701, Korea; chookh@knu.ac.kr

\* Correspondence: pongsak.n@ku.ac.th; Tel.: +66-2797-0999 (ext. 1008)

**Abstract:** At a low COD:TN ratio ($\leq$5) in influent, maintaining a longer HRT ($\geq$9 h) and longer SRT ($\geq$30 d) are suggested to improve higher N removal efficiency in case of operation at low DO (Dissolved oxygen) level ($0.9 \pm 0.2$ mg-$O_2$/L). However, in case of operation at high DO level ($4.0 \pm 0.5$ mg-$O_2$/L), short HRT (1 h) and typical SRT (17 d) make it possible to achieve nitrogen removal. On the other hand, at a high COD:TN ratio ($\geq$8.4), a typical HRT (9–15 h), SRT (12–19 d), and DO level (1.3–2.6 mg-$O_2$/L) would be applied. Microbial distribution analysis showed an abundance of AOA (Ammonia-oxidizing archaea) under conditions of low DO ($\leq$0.9 mg-$O_2$/L). *Nitrosomonas* sp. are mostly found in the all investigated water resource recovery facilities (WRRFs). *Nitrosospira* sp. are only found under operating conditions of longer SRT for WRRFs with a low COD:TN ratio. In comparison between abundances of *Nitrobacter* sp. and *Nitrospira* sp., abundances of *Nitrobacter* sp. are proportional to low DO concentration rather than abundance of *Nitrospira* sp. A predominance of *nosZ*-type denitrifiers were found at low DO level. Abundance of denitrifiers by using *nirS* genes showed an over-abundance of denitrifiers by using *nirK* genes at low and high COD:TN ratios.

**Keywords:** pre-anoxic; COD:TN; nitrogen removal; microbial communities

## 1. Introduction

Organic matter and inorganic nutrients (nitrogen, N and phosphorus, P) are the main contaminants to be treated in municipal wastewaters. Discharge of inorganic nutrients into the environment is responsible for eutrophication or algal blooms and toxic effects to aquatic life. For this reason, organic matter and inorganic nutrients from municipal wastewaters need to be removed before being discharged to our environment. A biological treatment process is often recommended because of its high removal efficiency and inexpensive operational costs compared to physical and chemical treatment processes. Pre-anoxic systems, which include Modified Ludzack–Ettinger (MLE), step feed, and sequencing batch reactor (SBR), are popular. These systems consist of an anoxic tank (first zone) followed by an aerobic tank (second zone) and are specifically designed for N removal. An anaerobic system prior to anoxic and aerobic systems is designed and operated biologically, in which there is an abundance of microorganisms responsible for both N and P removals. With alkalinity provided for the nitrification step (aerobic zone) and produced denitrification step (anoxic zone), N can be removed efficiently and a good settling sludge can be produced. The energy cost of this system is low, and operation is relatively simple. Moreover, internal nitrate recycling through proper control and return of activated sludge (RAS) from the

aerobic zone to the anoxic zone is the key to operating the process successfully [1]. When designing an anaerobic system prior to an anoxic and an aerobic system (or w/- and w/o anaerobic at front), wastewater characteristics, such as chemical oxygen demand (COD), total nitrogen (TN), and operational parameters including contact time in the anaerobic tank, the solids retention time (SRT), the hydraulic retention time (HRT), and the DO concentration must be taken into consideration. The proper COD:TN ratio in influent wastewater is an important parameter for biological N removal. In municipal wastewater with a low COD:TN ratio, there is insufficient carbon for the denitrification process, resulting in low N removal [2]. External carbon source addition is a significant approach to improve biological N removal (BNR) performance for wastewater with a low COD:TN ratio [3]. However, adding an external carbon source could be expensive in the case of a full-scale WRRF, where there is high capacity. To save costs, operating with longer SRT might be a potential approach to improve biological N removal performance for wastewater with a low COD:TN ratio. Phanwilai et al. [4] achieved significant N removal with a step feed treatment process operated at an SRT >60 d. Liu et al. [5] reported that a system with an SRT at 40 d outperformed systems with shorter SRTs (5, 10, and 20 d).

Maintaining low DO level in the aerobic tank could be another operating parameter to increase BNR performance. In instances with very low DO levels, such as 0–0.5 mg-$O_2$/L, ammonia-oxidizing archaea (AOA) would be the dominant microorganism group responsible for N removal [6]. Increasing the abundance of ammonia-oxidizing bacteria (AOB) was reported with a high DO level of 1.9–3.5 mg-$O_2$/L [6]. The domination of Nitrospira was observed at DO below 1.0 mg-$O_2$/L [7].

Temperature and free ammonia (FA) are also important factors affecting the microbial community. A range of temperature of 10–20 °C was reported to be optimal for Nitrospira [8] and a temperature of 24–25 °C is favorable for Nitrobacter [7]. FA was an inhibitor of nitrite-oxidizing bacteria (NOB) activity [9]. Furthermore, Nitrobacter is more sensitive to FA than Nitrospira [10].

Total nitrogen removal evidence for full-scale (pre-anoxic systems) municipal WRRFs, especially for low and high COD:TN ratios, longer and typical SRTs, and various DO concentrations and temperatures is not available. For this reason, this research focused on a comparison of N removal performance, and identification and quantification of microbial communities from anaerobic, anoxic, and aerobic tanks in four full-scale municipal WRRFs having low (≤5) to high (≥8.4) COD:TN ratios. In this work, only N removal efficiencies with operational parameters (HRT, SRT, and DO level) were discussed by using results from microbial abundance and communities of bacteria related to N removal, such as AOA, AOB, NOB, and denitrifying bacteria (DNB). In addition, the results from this work could be applied to increase N removal efficiencies of other pre-anoxic w/- and w/o anaerobic WRRFs that have low and high COD:TN ratios in influent.

## 2. Materials and Methods

### 2.1. Full Scale Descriptions

A total of four full-scale municipal wastewater treatment plants were investigated: two pre-anoxic without (w/o) anaerobic process, located at the Dindaeng water resource recovery facility (WRRF), Bangkok, Thailand (L1) and the Metro Wastewater Reclamation District (MWRD), Denver, USA (H1); and two pre-anoxic with (w/-) anaerobic processes, located at the Dalseocheon WRRFs, Daegu, South Korea (L2) and the Suvarnabhumi Airport WRRF, Samutprakarn, Thailand (H2). The full-scale WRRFs were mainly designed for biological nutrient removal (BNR), especially for removal of both N and P. The influent COD:TN mass ratios at L1, L2, H1, and H2 were 3.7, 4.2, 10.9, and 8.4, respectively. Low and high COD:TN ratios of WRRFs are ≤5 and ≥8.4, respectively. At H2, the wastewaters were mainly generated from aircrafts and business and commercial buildings, such as hotels and airlines' offices, in the area surrounding the Suvarnabhumi airport.

The schematic layouts of the full-scale pre-anoxic without (w/o) anaerobic processes at the L1 and H1 plants are shown in Figure 1A,B, respectively, and the full-scale pre-anoxic

with (w/-) anaerobic processes at L2 and H2 are shown in Figure 1C,D, respectively. No primary clarifier was designed for L1 or H2 in Thailand. A total of two internal recycles are designed in these plants: the first is from an aerobic zone to an anoxic zone and the second is for the return activated sludge (RAS) that is recycled from the 2nd clarifier back to the anaerobic system. All wastewater samples (*n* = 12 samples) from these four full-scale WRRFs were collected every month from each sampling point (anaerobic, anoxic, and aerobic zones) twice between 2018 and 2019 (before the COVID-19 pandemic).

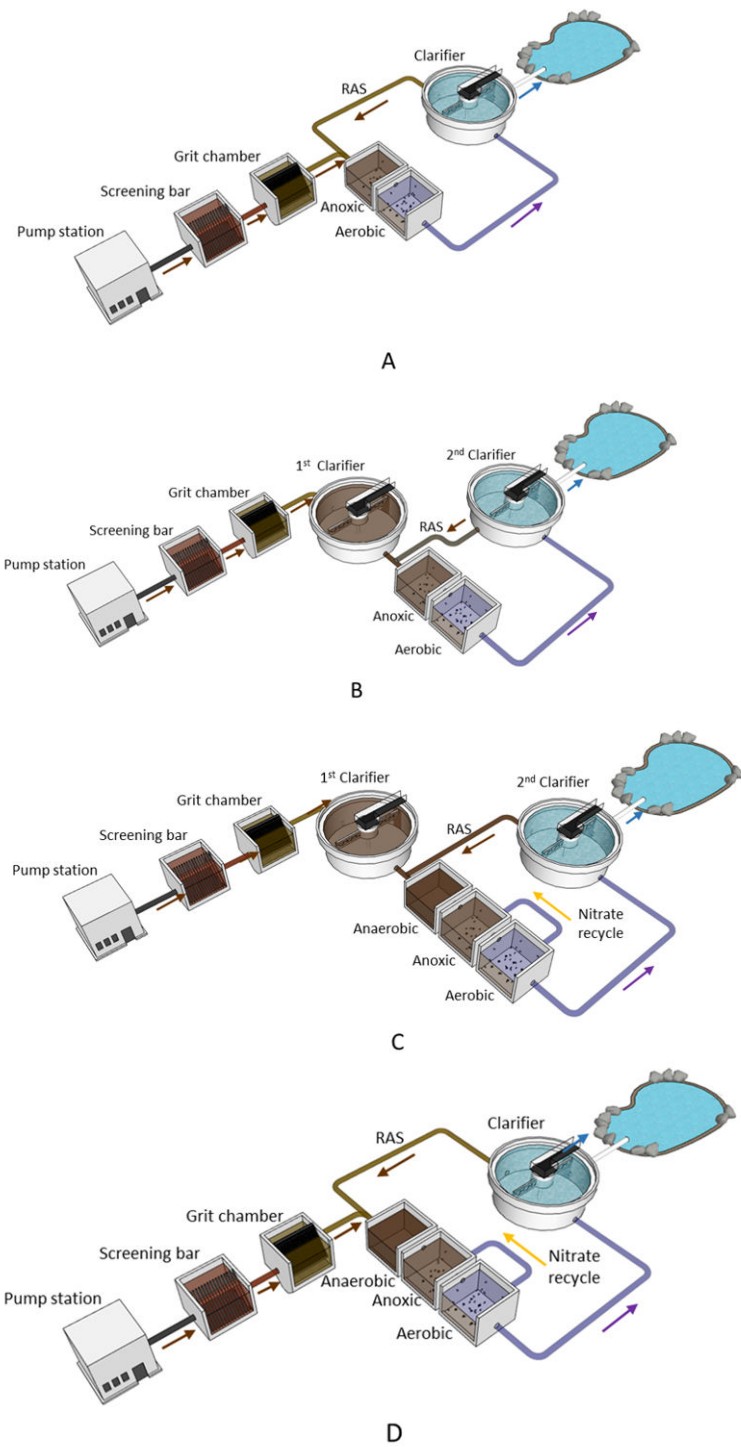

**Figure 1.** The schematic layout of the full-scale WRRFs at (**A**) L1, (**B**) H1, (**C**) L2, and (**D**) H2.

### 2.2. Wastewater Quality Analysis

BOD, COD, $NH_4^+$-N, $NO_2^-$-N, $NO_3^-$-N, organic-N, TKN, TN, TP, TSS, and SS from all wastewater samples were analyzed by following the standard method [11]. Only two effluent wastewater samples from L2 and H1 were measured for *E. coli*.

### 2.3. Microbiological Analysis

Molecular analysis of microbes was conducted on selected sludge samples from anaerobic, anoxic, and aerobic zones. Before the DNA extraction step, the sludge from each zone was harvested and kept on ice. A total of one mL of sludge was used for DNA extraction according to the procedures of Zhou et al. [12].

Focusing on microbial abundance by quantitative polymerase chain reaction (qPCR) analysis, a 20-µL sample was mixed with 1 µL of template DNA and 20 pmol of each primer. All qPCR reactions were performed by using a CFX96 Touch$^{TM}$ Real-Time PCR and CFX Manager version 3.1.1517.0823 (Bio Rad Laboratories, Inc., Hercules, CA, USA). Efficiency, slope, and $r^2$ values of individual real-time PCR assays are 98.3–106.1%, $(-3.1)$–$(-3.4)$, and 0.993–0.997, respectively, and the linearity range is $10^1$–$10^8$, see Table S1. Total bacteria were identified via 16S-rRNA EUB gene. Ammonium-oxidizing bacteria and archaea were identified through AOB-and AOA-*amoA* genes. NOB were identified as Nitrospira and Nitrobacter via 16S rRNA *NSR* and *Nitro* genes, respectively. DNB were identified via *nirS*, *nirK,* and *nosZ* genes. Pair oligonucleotide primers of EUB gene were performed with 338F/518R (5'-ACT CCT ACG GGA GGC AGC-3' [13]/5'-TAC CGC GGC TGC TGG CAC-3' [14]), *amoA* gene of AOB with amoA-1F/2R (5'-GGG GTT TCT ACT GGT GGT-3'/5'-CCC CTC KGS AAA GCC TTC TTC–3') [15], AOA with Arch-amoAF/AR (5'-STA ATG GTC TGG CTT AGA CG-3'/5'-GCG GCC ATC CAT CTG TAT GT-3') [16], 16S rDNA-Nitrobacter with Nb1000F/1387R (5'-TGC GAC CGG TCA TGG-3'/5'-GGG CGG WGT GTA CAA GGC-3') [17], 16S rDNA-Nitrospira with NSR1113F/1264R (5'-CCT GCT TTC AGT TGC TAC CG-3' [17]/5'-GTT TGC AGC GCT TTG TAC CG-3' [18]), DNB genes of *nirS* gene with cd3AF/R3cd (5'-GTS AAC GTS AAG GAR ACS GG-3' [19]/5'-GAS TTC GGR TGS GTC TTG A-3' [20]), *nirK* gene with F1aCu/ R3Cu (5'-ATY GGC GGV CAY GGC GA-3'/5'-GCC TCG ATC AGR TTR TGG TT-3') [21], *nosZ* gene with nosZ2F/2R (5'-CGC RAC GGC AAS AAG GTS MSS GT-3'/5'-CAK RTG CAK SGC RTG GCA GAA-3') [22].

Microbial communities responsible for N removal were determined by Denaturing gradient gel electrophoresis (DGGE) analysis. Pair oligonucleotide primers of 16S rRNA AOB gene were performed by nested PCR protocol with 2 steps; 1st step: CTO189fABC/CTO654r (5'-GGA GRA AAG YAG GGG ATC G-3'/5'-CTA GCY TTG TAG TTT CAA ACG C-3') [23], and 2nd step: 357f-GC/518r (5'-CCT ACG GGA GGC AGC AG-3'/5'-ATT ACC GCG GCT GCT GG-3') [14]. DNB genes were performed with *nirS* gene with cd3AF/R3cd-GC (5'-GTS AAC GTS AAG GAR ACS GG-3'/5'-GAS TTC GGR TGS GTC TTG A–3') [20], and *nirK* gene with F1aCu/R3Cu-GC (5'-ATY GGC GGV CAY GGC GA-3'/5'-GCC TCG ATC AGR TTR TGG TT-3') [20]. Each 25-µL reaction mixture was added to 1 µL of template DNA with concentrations of 10–20 ng/µL 10× *Ex Taq*$^{TM}$ buffer, 5 units/µL *TaKaRa Ex Taq*$^{TM}$, 2.5 mM dNTP Mixture, and 10 pmol of each primer, and the mixture was finally diluted with nuclease-free water. All PCR reactions were performed by using a T100$^{TM}$ Thermal cycler (BioRad Laboratories, Hercules, CA, USA). The PCR product of 15 µL was loaded into individual lanes on 8% (*w/v*) acrylamide gel with 35–55% gradient for EUB target and with 35–50% gradient for AOB target. The electrophoresis step was performed in 1× TAE buffer at 58 °C with a constant voltage of 80 V for 16 h. The shaped DNA band on acrylamide gel was excised by a scalpel. The DNA fragments were eluted by milli-Q water and set aside in a refrigerator overnight, and then amplified by PCR with the same primer without attached CG-camp. Sequencing bases were aligned by using database of the National Center for Biotechnology Information (NCBI).

*2.4. Calculations*

The removal efficiencies (%) of nutrients and contaminants were calculated using Equation (1), where $C_{inf}$ and $C_{out}$ are concentrations (mg/L) of water quality parameters in influent and effluent of a treatment process, respectively.

$$\text{Removal efficiency (\%)} = \frac{C_{inf} - C_{out}}{C_{inf}} \times 100 \tag{1}$$

COD loading rate (kg COD/m$^3$·d), BOD loading rate (kg BOD/m$^3$·d), and ammonia loading rate (ALR) (kg NH$_4^+$-N/m$^3$·d) were calculated according to Equations (2)–(4), respectively, where TCOD$_{inf}$ and BOD$_{inf}$ are concentration (mg/L) of total COD of the influent, (kg COD/m$^3$) and BOD concentration of the influent, (kg BOD/m$^3$), respectively. NH$_4^+{}_{inf}$ is the ammonia concentration of the influent, (kg NH$_4^+$-N/m$^3$), Q is flow rate, (m$^3$/d), and V is volume of the reactor, (m$^3$).

$$\text{TCOD } (\text{kg} - \text{N}/m^3 \cdot \text{d}) = \frac{\text{TCOD}_{inf} \times Q}{V} \tag{2}$$

$$\text{BOD } (\text{kg} - \text{N}/m^3 \cdot \text{d}) = \frac{\text{BOD}_{inf} \times Q}{V} \tag{3}$$

$$\text{ALR } (\text{kg} - \text{N}/m^3 \cdot \text{d}) = \frac{\text{NH}_4^+{}_{inf} \times Q}{V} \tag{4}$$

Free ammonia (FA) was calculated using Equation (5) according to Anthonisen et al. [24], where NH$_4^+{}_{inf}$ is the influent ammonium concentration (mg-N/L) and T is the temperature of the effluent (°C).

$$\text{FA } (\text{mg} - \text{N}/\text{L}) = \frac{17}{14} \times \frac{[\text{NH}_4^+]_{inf} \times 10^{\text{pH}}}{\exp\left[\frac{6334}{273+\text{T}} + 10^{\text{pH}}\right]} \tag{5}$$

*2.5. Statistical Analysis for Microbial Abundances*

One-way analysis of variance (one-way ANOVA) with Tukey's honestly significant difference (HSD, at $p < 0.05$) was performed using Minitab 18.1 for microorganism abundance as copies-DNA. The level for statistical significance was 95%.

**3. Results and Discussion**

*3.1. Major Operational Parameters and Performance of Full-Scale Pre-Anoxic w/o and w/-Anaerobic Process*

The operational parameters (SRT, HRT, and DO) of the four WRRFs are presented in Table 1. Comparing these operational parameters at L1, L2, H1, and H2, low DO level in aerobic zone (0.9 ± 0.2 mg-O$_2$/L), longer SRT of 30 d, and HRT (8 h) were found at L1 and high DO level (4.0 ± 0.5 mg-O$_2$/L), and shorter SRT of 17 d and HRT (3.6 h) were found at L2. At H1, a quite low DO level (1.3 ± 0.4 mg-O$_2$/L), SRT of 12 d, and longer HRT (9.5 h) were found, while a quite high DO level (2.6 ± 0.2 mg-O$_2$/L), typical SRT of 19 d, and longer HRT (15.4 h) were found at H2.

**Table 1.** Operational parameters of the full-scale pre-anoxic zone w/- and w/o anaerobic systems by low and high COD:TN ratio.

| Operational Parameter | Low COD:TN ($\leq$5) | | High COD:TN ($\geq$8.4) | |
|---|---|---|---|---|
| | **L1 (w/o)** | **L2 (w/-)** | **H1 (w/o)** | **H2 (w/-)** |
| SRT (d) | 30 | 17 | 12 | 19 |
| HRT (total) (h) | 7.5 | 3.6 | 9.5 | 15.4 |
| Anaerobic | - | 1.0 | - | 1.3 |
| Anoxic | 1.5 | 1.6 | 1.5 | 3.1 |
| Aerobic | 6.0 | 1.0 | 8.0 | 11.0 |
| DO (mg-$O_2$/L) | | | | |
| Anoxic | $0.3 \pm 0.1$ | Negligible | Negligible | 0.1 |
| Aerobic | $0.9 \pm 0.2$ | $4.0 \pm 0.5$ | $1.3 \pm 0.4$ | $2.6 \pm 0.2$ |

The average physical and chemical characteristics of wastewater quality of the four full-scale cases are compared in Table 2. The average flow rates at L1 and L2 were high compared to H1 and H2. BOD (30 mg/L) and (75 mg/L) were found at L1 and L2, respectively. BOD (283 mg/L) and (260 mg/L) were found at H1 and H2, respectively. L1 and L2 received wastewaters with BOD (from 30 mg/L to 75 mg/L) because they treated wastewater collected from a combined sewer system with domestic sewage being diluted by storm water. Infiltration and inflow are able to enter this combined sewer system. Additionally, at L1, the high temperature inside the sewer lines could promote the degradation of BOD, and septic tank installation in the residential houses could remove BOD before wastewater entering the sewer lines. The H1 and H2 WRRFs received wastewaters with high BOD. At these WRRFs, sewage and storm water lines are separated.

BOD, COD, and N removal efficiencies in the four full-scale cases are shown at the bottom of Table 2. At low COD:TN, BOD, COD, $NH_4^+$-N, and TN removal efficiencies were 83%, 67%, 95%, and 49%, respectively at L1 and, 96%, 89%, 99%, and 70%, respectively, at L2. At high COD:TN, BOD, COD, $NH_4^+$-N, and TN removal efficiencies were 98%, 98%, 91%, and 81%, respectively at H1, and 98%, 92%, 91%, and 86%, respectively, at H2.

The average N concentration and removal efficiencies in each month are shown in Figure 2. The total nitrogen (TN) removal efficiency at L1 was quite low (only 49%) in comparison to the other WRRFs. The low TN removal could be explained by the low COD:TN ratio ($\leq$5) in the wastewater received at L1. Low N removal efficiencies were also reported by Liu et al. [25] for WRRFs treating wastewater of relatively low COD:TN ratios. It was reported that the denitrification process could not significantly occur due to the insufficient carbon source for denitrification in wastewater having relatively low COD:TN ratios. On the contrary, the L2 with COD:TN ratio of 4.2 had an efficient TN removal of 70%. It is postulated that the plant operator has to operate with high DO level ($4.0 \pm 0.5$ mg-$O_2$/L), short HRT (1 h), and typical SRT (15–20 d). Associated with typical SRT, the plant operator really needs to keep significantly low or negligible DO concentration in the anoxic tank for the denitrification process to occur. In this case, it requires skillful operators to control the system correctly.

**Table 2.** Comparing the average physical and chemical characteristics of wastewater quality in full-scale pre-anoxic zone w/- and w/o anaerobic systems by low and high COD:TN ratio.

| Parameter | Low COD:TN ($\leq$5) | | High COD:TN ($\geq$8.4) | |
|---|---|---|---|---|
| | L1 (w/o) | L2 (w/-) | H1 (w/o) | H2 (w/-) |
| **Inlet/Outlet** | | | | |
| pH | $7.2 \pm 0.01/7.2 \pm 0.01$ | $7.2 \pm 0.2/6.9 \pm 0.2$ | $7.0 \pm 0.2/7.2 \pm 0.1$ | $7.2 \pm 0.1/7.2 \pm 0.03$ |
| Temp (°C) | $28.1 \pm 0.5/27.7 \pm 0.4$ | $21.5 \pm 2.5/24 \pm 0.7$ | $18.5 \pm 2.3/18.5 \pm 0.6$ | $27 \pm 0.1/26.9 \pm 0.3$ |
| SS (mg/L) | $46.7 \pm 5.8/8.6 \pm 0.7$ | $202 \pm 71.3/2 \pm 0.3$ | $321.7 \pm 122.5/15.3 \pm 1.3$ | $178.5 \pm 38.1/4.7 \pm 2.5$ |
| BOD (mg/L) | $30.1 \pm 2.7/5.0 \pm 1.2$ | $75 \pm 26.4/3 \pm 0.3$ | $283.0 \pm 43.2/4.4 \pm 0.7$ | $260 \pm 6.52/1 \pm 0.3$ |
| COD (mg/L) | $58 \pm 25.9/19 \pm 1.5$ | $88 \pm 25.1/10 \pm 0.6$ | $452.8 \pm 48.5/7.4 \pm 1.8$ | $511.6 \pm 36.0/40.5 \pm 1.3$ |
| $NH_4^+$ (mg-N/L) | $11.0 \pm 1.1/0.6 \pm 0.2$ | $10.6 \pm 2.2/0.2 \pm 0.1$ | $28.5 \pm 2.3/0.5 \pm 0.2$ | $55.4 \pm 7.3/4.8 \pm 0.2$ |
| $NO_3^-$ (mg-N/L) | $0.2 \pm 0.07/5.3 \pm 1.0$ | $0.1 \pm 0.02/6.3 \pm 2.5$ | -/- | -/- |
| Alkalinity (mg/L) | -/- | -/- | $247.8 \pm 8.0/126.4 \pm 7.8$ | $344.2 \pm 9.3/154.3 \pm 45.5$ |
| TKN (mg/L) | $15.4 \pm 1.4/2.6 \pm 0.9$ | -/- | $46.2 \pm 5.6/-$ | $60.8 \pm 6.6/6.1 \pm 0.3$ |
| TN (mg-N/L) | $15.6 \pm 1.4/8.0 \pm 0.3$ | $27 \pm 5.1/8 \pm 0.4$ | $41.3 \pm 2.9/7.9 \pm 0.7$ | $61.2 \pm 6.9/8.9 \pm 0.1$ |
| TP (mg-P/L) | $2.3 \pm 0.1/1.5 \pm 0.2$ | $4 \pm 1.1/0.2 \pm 0.02$ | -/- | $7.1 \pm 0.2/0.3 \pm 0.1$ |
| *E. Coli* (MPN) | -/- | $44,845 \pm 20,782.6/22 \pm 42.8$ | -/40 | -/- |
| **Removal Efficiency (%)** | | | | |
| SS | 82 | 99 | 95 | 97 |
| BOD | 83 | 96 | 98 | 98 |
| COD | 67 | 89 | 98 | 92 |
| $NH_4^+$ | 95 | 99 | 98 | 91 |
| TKN | 83 | - | - | 90 |
| TN | 49 | 70 | 81 | 86 |
| TP | 35 | 95 | - | 96 |
| **Other information** | | | | |
| Avg. Flow rate (m$^3$/d) | 218,433 | 220,655 | 77,917 | 7673 |
| MlSS (mg/L) | $4509.5 \pm 414.17$ | $3455 \pm 380$ | $3577 \pm 515$ | $3315 \pm 328$ |
| MLVSS (mg/L) | $2542 \pm 414$ | $2780 \pm 280$ | $2862 \pm 412$ | $2610 \pm 208$ |
| COD:TN ratio | 3.7 | 4.2 | 10.9 | 8.4 |
| COD loading rate (kg-COD/m$^3$·d) | 0.19 | 0.59 | 0.71 | 1.29 |
| BOD loading rate (kg-BOD/m$^3$·d) | 0.10 | 0.50 | 0.44 | 0.50 |
| ALR (kg $NH_4$ N/m$^3$·d) | 0.04 | 0.07 | 0.04 | 0.14 |
| TNLR (kg-N/L-d) | 0.05 | 0.18 | 0.06 | 0.15 |
| TNRR (kg-N/L-d) | 0.02 | 0.25 | 0.05 | 0.13 |
| FA (mg-N/L) | 0.15 | 0.17 | 0.15 | 0.28 |

Remark: - = Not record.

For WRRFs treating COD:TN ratio (<4) wastewater, maintaining a very long SRT ($\geq$60 d) is recommended to overcome the low TN removal efficiency [4] as the longer SRT would increase the nitrifying bacteria abundance. Meanwhile, a long SRT could also enhance $NH_4^+$ removal by increasing nitrification activity [1]. The effluent $NH_4^+$ concentration at H2 was 4.8 mg-N/L and this was the highest among the WRRFs studied due to the plant having the highest $NH_4^+$ concentration in the raw water (55.4 mg-N/L). The effluent $NH_4^+$ concentrations in the activated sludge process were reported for SRTs at 5 d ($2.6 \pm 2.3$ mg-N/L), 10 d ($0.04 \pm 0.01$ mg-N/L), 20 d ($0.03 \pm 0.007$ mg-N/L), and 40 d ($0.02 \pm 0.003$ mg-N/L), corresponding to $NH_4^+$-N removal efficiencies of 94.5%, 99.9%, 99.9%, and 99.9%, respectively [5]. To further enhance the removal of $NH_4^+$-N, a long SRT of >19 d is recommended because it is assumed that a complete biodegradation of organic matters including readily biodegradable COD (rbCOD) and slow biodegradable COD (sbCOD) and endogenous decay of bacteria could occur due to long SRT conditions, which significantly affects denitrification process.

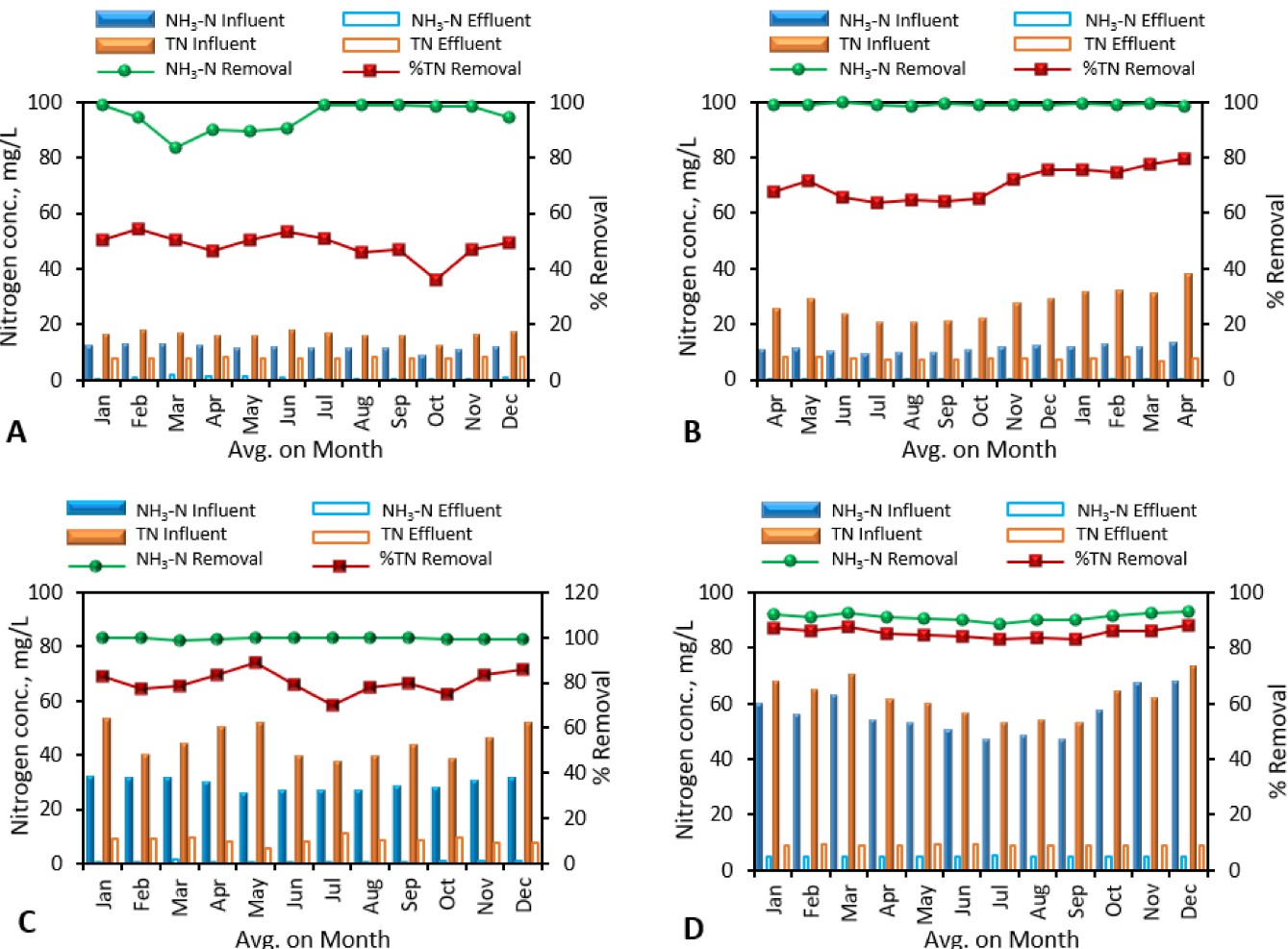

**Figure 2.** Nitrogen performance with low COD:TN ratio at (**A**) L1 and (**B**) L2, and high COD:TN ratio at (**C**) H1 and (**D**) H2.

To solve this carbon limitation at L1 WRRF without external carbon addition implies that a future operator could operate a system with HRT ($\geq$9 h) and long SRT ($\geq$30 d). Associated with longer SRT, a low DO level (0.9 $\pm$ 0.2 mg-$O_2$/L) is able to be maintained in aerobic tank. It could be postulated that the reason to remain under all these conditions is that partial nitritation and simultaneous nitrification and denitrification (SND) processes are expected to occur in the aerobic tank; possible evidence to support this statement is in Section 3.2. Furthermore, nitrogen-cycling microbial abundances and communities are related to the various environmental factors such as DO level, SRT, temperature, pH, and ammonium loading rates (ALRs), etc.

### 3.2. Nitrogen-Cycling Microbial Abundances and Communities
3.2.1. Ammonia-Oxidizing Archaea (AOA) and Ammonia-Oxidizing Bacteria (AOB) Targeting

Autotrophic nitrifying bacteria responsible for ammonia oxidation process were detected at L1 and belonged to two orders: Nitrosomonadales (affiliated with *Nitrosomonas* sp. *Nitrosospira* sp., *Nitrosococcus* sp., and *Thiobacillus* sp.) and Rhodocyclales (affiliated with Azospira sp., *Thauera* sp., and *Zoogloea* sp.) as shown in Table 3. Zhang et al. [26] reported that in full-scale municipal WRRFs, the most important genera of AOB were *Nitrosomonas* and *Nitrosospira*. Furthermore, they mentioned that *Nitrosomonas* were predominant. Consistently, in the full-scale w/- and w/o pre-anaerobic WRRFs, *Nitrosomonas* sp. are the most dominant AOB in the WRRFs operated at low and high DO levels. The

microbial community of *Nitrosospira* sp. was found at the L1 plant because this WRRF was operated under a long SRT, a favorable condition for the growth of *Nitrosospira* sp. (see Table 3). Although the abundance of *Nitrosospira* sp. is less than that of *Nitrosomonas* sp., the existence of *Nitrosospira* sp. might be a suitable factor for satisfying an efficient nitrification process when the conditions are not optimal for growth of nitrifying bacteria [27].

Figure 3A shows the abundance of AOA-*amoA* genes at the L1 and H1 w/o pre-anaerobic systems, which is higher than in the L2 and H2 w/- pre-anaerobic systems. The abundance at L1 is the highest among the full-scale WRRFs and the statistical significance of each zone shows the high mean difference of letter grouping (Table S2, identified *a* letter of anoxic and aerobic zones but the others show *b*, *c*, *cd*, and *d* letters, $p < 0.05$).

**Table 3.** Microorganisms' community in four municipal WRRFs.

| Order | Species | % | Accession No. | Low COD:TN (≤5) | | | | | High COD:TN (≥8.4) | | | | |
|---|---|---|---|---|---|---|---|---|---|---|---|---|---|
| | | | | L1 | | L2 | | | H1 | | H2 | | |
| | | | | Anx | Aer | Ana | Anx | Aer | Anx | Aer | Ana | Ana | Aer |
| **Nitrifying bacteria: Ammonia oxidizing bacteria (AOB)** | | | | | | | | | | | | | |
| Nitrosomonadale | *Nitrosomonasaestuarii* | 90 | NR104818.1 | × | × | | | | × | × | | | |
| | *Nitrosomonas eutropha* | 93 | NR027566.1 | × | × | × | × | × | | | | | |
| | *Nitrosomonas communis* | 97 | NR119314.1 | × | × | × | × | × | | | × | × | × |
| | *Nitrosomonas halophila* | 93 | NR104817.1 | × | × | × | × | × | × | × | | | |
| | *Nitrosomonas marina* | 99 | NR104815.1 | | | | | | × | × | | | |
| | *Nitrosomonas oligotropha* | 96 | NR104820.1 | × | × | | | | | | × | × | × |
| | *Nitrosomonas stercoris* | 98 | NR146824.1 | | | | | | × | × | | | |
| | *Nitrosomonas ureae* | 97 | NR104814.1 | × | × | | | | | | × | × | × |
| | *Nitrosospira multiformis* | 96 | NR074736.1 | × | × | | | | | | | | |
| | *Nitrosospira tenuis* | 97 | NR114773.1 | × | × | | | | | | | | |
| | *Uncultured Nitrosospira* | 95 | GQ255611.1 | × | × | | | | | | | | |
| | *Thiobacillus thioparus* | 96 | NR117864.1 | × | × | | | | | | × | × | × |
| Rhodocyclales | *Zoogloea caeni* | 91 | NR043795.1 | | | | | | | | × | × | × |
| **Nitrifying bacteria: Nitrite oxidizing bacteria (NOB)** | | | | | | | | | | | | | |
| Nitrospirae | *Nitrospira lenta* | 99 | NR148573.1 | × | × | | | | | | | | |
| **Heterotrophic nitrifying bacteria (HNB)** | | | | | | | | | | | | | |
| Pseudomonadales | *Pseudomonas asturiensis* | 98 | NR108461.1 | | | | | | × | × | | | |
| | *Pseudomonas fragi* | 98 | MT176180.1 | | | | | | × | × | | | |
| | *Pseudomonas fluorescens* | 98 | CP027561.1 | | | | | | × | | | | |
| | *Pseudomonasputida* | 99 | MH778047.1 | | | | | | × | × | | | |
| **Denitrifying bacteria (DNB): Autotrophic denitrifying bacteria** | | | | | | | | | | | | | |
| Chloroflexi | *Chloroflexi bacterium* | 87 | KP246879.1 | | | | | | × | | | | |
| | *Uncultured Chloroflexi* | 98 | GQ366686.1 | | | × | × | × | | | | | |
| Rhodocyclales | *Azospira restricta* | 97 | NR044023.1 | × | × | | | | | | | | |
| | *Thauera aromatica* | 100 | NR026153.1 | × | | | | | | | | | |
| | *Thauera aminoaromatica* | 93 | NR027211.1 | | | | | | | × | | | |
| Saprospirales | *Haliscomenobacter hydrossis* | 90 | NR074420.1 | × | × | × | × | × | | | × | × | × |
| **Denitrifying bacteria (DNB): Heterotrophic denitrifying bacteria** | | | | | | | | | | | | | |
| Acidimicrobiales | *Ilumatobacter fluminis* | 86 | NR041633.1 | | | × | × | × | | | | | |
| Burkholderiales | *Comamonas denitrificans* | 99 | NR025080.1 | | | | | | × | | | | |
| | *Comamonas phosphati* | 96 | NR147778.1 | | | | | | × | | | | |
| | *Rhodoferax ferrireducens* | 92 | NR074760.1 | | | × | × | × | | | × | × | × |
| Chitinophagales | *Terrimonas lutea* | 96 | NR041250.1 | | | × | × | × | | | × | × | × |
| | *Niabella terrae* | 92 | NR132698.1 | | | | | | | | | × | × |
| | *Sediminibacterium roseum* | 82 | NR159130.1 | | | × | × | × | | | | | |
| Rhodospirillales | *Tistrella mobilis* | 91 | NR117256.1 | × | × | | | | | | | | |
| Micrococcales | *Oryzobacter terrae* | 98 | NR137270.1 | × | × | × | × | × | | | | | |

Remark: × is DGGE band presenting on acrylamide gel, Ana is an anaerobic system, Anx is an anoxic zone, and Aer is an aerobic zone.

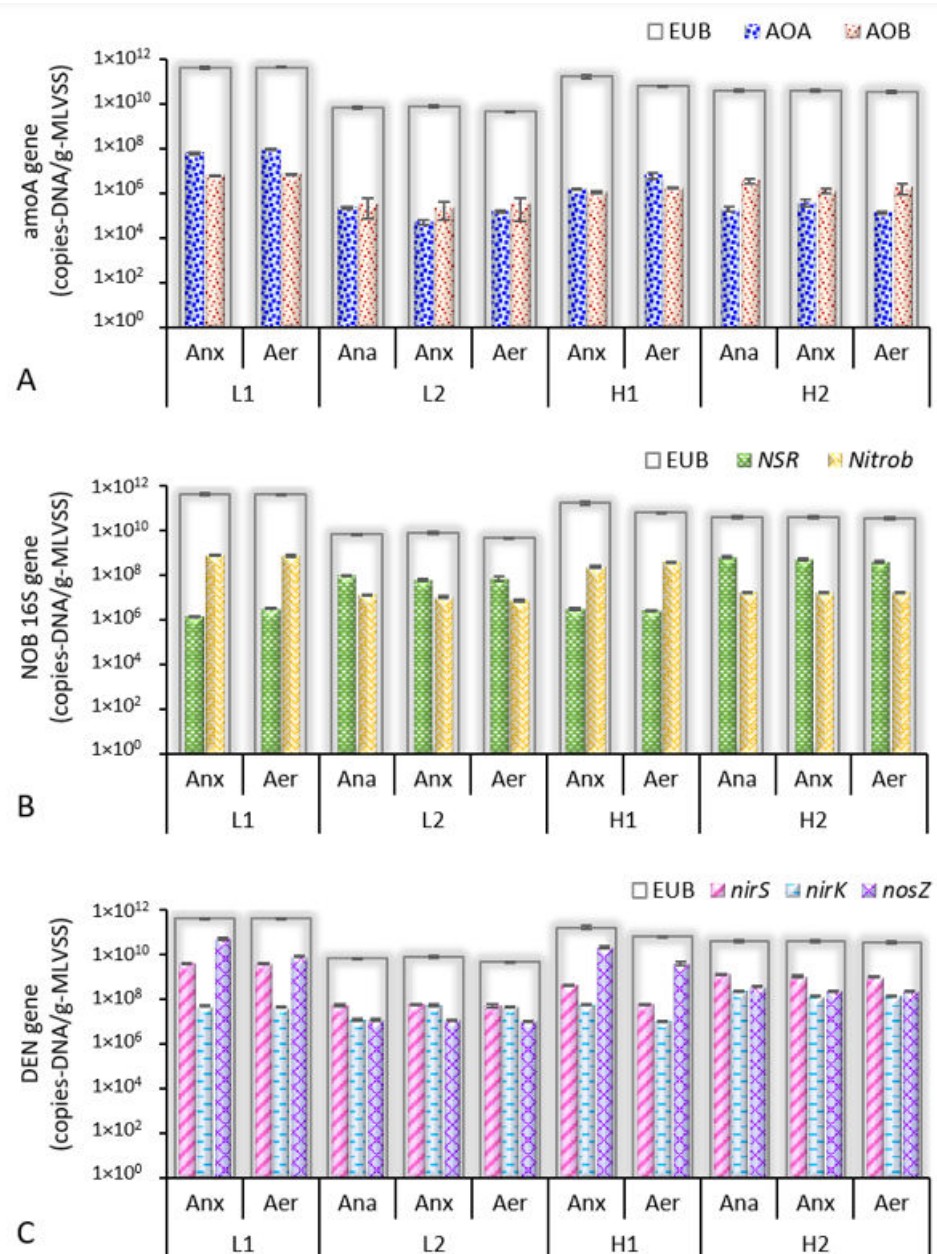

**Figure 3.** Microbial abundance of (**A**) *amoA* gene-AOA and -AOB, (**B**) NOB, and (**C**) DEN at L1, L2, H1, and H2.

Both the L1 and H1 w/o pre-anaerobic systems had higher AOA abundance, which was expected because the lower DO level, higher temperature, and longer SRT (>30 d) would significantly promote the growth of AOA. This result is similar to the result by Yin et al. [28]. Gao et al. [6] studied the effects of DO levels on the growth of AOB-*amoA* and AOA-*amoA*, showing the former is more abundant under high DO levels of 1.9–3.5 mg-$O_2$/L. Phanwilai et al. [4] analyzed the abundance of microorganisms in the step-feed aerobic tanks of a municipal WRRF, reporting that AOA-*amoA* were the most abundant genes in the tank with low DO levels ($0.9 \pm 0.5$ mg-$O_2$/L), while AOB-*amoA* genes were higher than AOA-*amoA* genes in the tank with high DO level ($1.8 \pm 0.5$ mg-$O_2$/L). In this work, the result of AOB and AOA abundance at L2 and H2 WRRFs, which are operated at high DO levels of 2.4–4.5 mg-$O_2$/L, are in line with the results by Gao et al. [6] and Phanwilai et al. [4] (see Figure 3A). Other factors such as the high $NH_4^+$ loading rate could also increase AOB abundance. The predominance of the AOB-*amoA* gene over the

AOA-*amoA* gene at L2 and H2 compared to L1 and H1 could be attributed to the higher $NH_4^+$ loading rates in those plants (see Table 2), and the significance of the gene ($p$ <0.05) shown by the difference of letter grouping (see Table S2). The typical design DO level for a nitrogen-removal process of around 2 mg-$O_2$/L was recommended by [1].

Although an abundance of AOA was not found at the L2 and H2 WRRFs, AOA and AOB would collaborate and offer a possible advantage in ammonia oxidation rates at the lower ammonia concentration at L1 and H1. It is postulated that in the practical operation, it is desired to maintain low DO level in an aerobic tank to reduce energy and sustain SRT range based on characteristics of each full-scale WRRF, and the abundance of AOA might be a possible group of microorganisms to collaborate with AOB for the nitrification process. However, in further research a suitable DO level and SRT range would be investigated to find the optimum conditions of growth of AOA that could collaborate with AOB.

### 3.2.2. Nitrite-Oxidizing Bacteria (NOB) Targeting

Figure 3B shows that *Nitrobacter* was more abundant than *Nitrospira* at L1. The DO levels (0.7 to 1.1 mg-$O_2$/L) at L1 are the lowest among the WRRFs investigated; H1 was 0.9 to 1.7 mg-$O_2$/L and the DO concentration ranged from 2.4 to 4.5 mg-$O_2$/L for the other two WRRFs. The low DO condition is favorable for the growth of *Nitrobacter* and presented the highest significance in each zone, their grouping showed they were statistically different w/out all the extra text on the letters themselves ($p$ < 0.05, Table S2). However, Huang et al. [7] reported that DO concentration of >1.0 mg-$O_2$/L was a suitable condition for the growth of *Nitrobacter,* while a DO concentration of <1.0 mg-$O_2$/L was optimum for growth of *Nitrospira*. Similarly, Park et al. [29] suggested that at the low operational DO concentration of 0.5–0.6 mg-$O_2$/L, *Nitrospira* was selectively enriched over *Nitrobacter* in the activated sludge from a small-scale SBR. Furthermore, Liu and Wang [30] investigated the nitrification performance of activated sludge with the long-term effect of low DO concentration, finding a higher abundance of *Nitrospira* ($10^{12}$) than *Nitrobacter* ($10^{10.4}$) under the condition of 0.16 mg-$O_2$/L.

Longer SRT might be possible to increase abundance of *Nitrospira*. Roots et al. [31] mentioned that *Nitrospira* increased from 3.1 to 53% under the DO level of 0.2–1.0 mg-$O_2$/L with a 99 d SRT and $NH_4^+$ 0–14 mg-N/L. Qian et al. [32] found *Nitrospira* decreased from 0.44% to 0.04% with a DO level of 0.8–1.5 mg-$O_2$/L with SRTs between 33 and 56 d and $NH_4^+$ 105 mg-N/L. Comparatively, Sun et al. [33] set a short SRT of 15 d with a DO concentration at 1.0 and 2.0 mg-$O_2$/L that *Nitrospira* increased 1.81 and 2.99%, respectively. Under the longer SRT (30 d) and DO level (0.7–1.1 mg-$O_2$/L) at L1 there was lower abundance of *Nitrospira* than *Nitrobacter*, while the three plants with the shorter SRT (17 to 26 d) and higher DO level (2.4–4.5 mg-$O_2$/L) presented higher abundance of *Nitrobacter* than *Nitrospira*. At L2 and H2, *Nitrospira* was more abundant than *Nitrobacter*. These plants were operated at DO concentrations of 2.4–4.5 mg-$O_2$/L, HRTs of 3.6 to 15.4 h, and SRTs of 17–19 d. These operational parameters along with the ammonium loading rate (ALR) of 0.07 and 0.14 $NH_4^+$-N/m$^3$·d, respectively, were important factors affecting *Nitrospira* growth but had a lesser effect on *Nitrobacter* growth. However, SRT might not be the sole major effect on *Nitrospira* but other factors: DO, temperature, $NH_4^+$ influent, pH, HRT, FA, and ALR could also be significant factors affecting the competition between *Nitrospira* and *Nitrobacter* [9].

During the collection of all samples of this work, the temperature was recorded from 18.5 to 28 °C. For this reason, the optimal temperature ranges for *Nitrobacter and Nitrospira* growth are not exactly reported. Huang et al. [7] concluded that *Nitrobacter* was the favorable species under the temperature ranges of 24–25 °C while *Nitrospira* dominated at a relatively high temperature range of 29–30 °C. On the contrary, Alawi et al. [34] indicated that a lower temperature range of 10–20 °C was the optimum condition for *Nitrospira* growth.

Meanwhile, *Nitrobacter* is more sensitive to free ammonia (FA) concentration compared to *Nitrospira* [10]. Mehrani et al. [9] reported that FA was a major inhibitor of NOB activity.

FA concentrations at L2 (0.17 mg-N/L) and H2 (0.28 mg-N/L) were higher than at L1 (0.15 mg-N/L) and H1 (0.15 mg-N/L). It could be postulated that the FA concentration was an inhibitor and decreased the abundance of *Nitrobacter* in these WRRFs w/- the anaerobic system, which have lower FA concentrations than L1 and H1.

In this work, only the qPCR technique was used to identify both *Nitrobacter* and *Nitrosipra*; using the specific primers to detect nitrifying bacteria population for *Nitrobacter* and *Nitrospira* are recommended in the further research. This is because *Nitrospira* are able to complete oxidation of $NH_4^+$ direct to $NO_3^-$ without conversion to $NO_2^-$ (complete ammonia oxidizer (comammox) process). If the information of *Nitrospira* in full-scale WRRF is reliable, a new approach for the comammox process would be applied for increasing biological N removal in the future.

### 3.2.3. Denitrifying Bacteria (DNB) Targeting

A total of three coding genes for nitrite (*nirK* or *nirS*) and nitrous oxide (*nosZ*) reductases were evaluated for the abundance of denitrifying bacteria from the four full-scale WRRFs. As indicated in Figure 3C, a higher abundance of *nosZ*-type denitrifiers was found at L1 among the WRRFs investigated due to the low COD:TN ratio of $\leq 3.7$ in L1 (see Table S2). The effects of the COD:TN ratio on the abundance of *nosZ*-type denitrifiers were consistent with the results reported by Yuan et al. [35] who reported that the abundance of *nosZ*-type denitrifiers was two orders of magnitude higher at an influent COD:TN ratio of 4.6 ($1.29 \times 10^8$ copies/g-SS) compared to an influent COD:TN ratio of 8.4 ($1.31 \times 10^6$ copies/g-SS) at the Beijing municipal WRRF in China.

The average number of DNB copies presenting at L1 and H1 shows that *nosZ*-type denitrifiers were predominant in anoxic and anaerobic zones. Wang et al. [36] found that the abundance of *nosZ* was a good indicator for rechecking oxygen levels of anoxic and anaerobic tanks. Based on this result, it can be concluded that the DO level in the anoxic and anaerobic tanks of L1 was quite low, and denitrifying bacteria could not use $NO_3^-$ as the electron acceptor for the denitrification process, resulting in poor denitrification efficiency at L1 in the anoxic condition. As shown in Table 1, the DO level in the anoxic zone at L1 was $0.3 \pm 0.1$ mg-$O_2$/L and the DO level of anoxic zone at H1 was negligible. It should be noted that the low denitrification efficiency at L1 could also be attributed to the low COD:TN ratio.

Tallec et al. [37] and Jia et al. [38] indicated that a low DO concentration in WRRFs favors nitrous oxide ($N_2O$) production during the nitrification and denitrification process. High abundance of *nosZ* gene in denitrifiers was also found in the aerobic tanks of L1 and H1 WRRFs. Henry et al. [22] indicated that *nosZ*-type denitrifiers could be responsible in $N_2O$ production. It could be postulated that the BNR process at L1 and H1 could produce higher $N_2O$ gas among WRRFs investigated due to the low DO levels of their plants ($0.9 \pm 0.2$ and $1.3 \pm 0.4$ mg-$O_2$/L, respectively).

On the other hand, a high abundance of *nirS*-type denitrifiers and lower abundance of *nosZ*-type denitrifiers were found in the anaerobic and anoxic zones at L2 and H2 due to high DO concentration (2.4–4.5 mg-$O_2$/L) operated by the pre-anoxic process w/- anaerobic system. Meanwhile, *nirS*-type denitrifiers were more prevalent than the *nirK*-type denitrifiers at all full-scale WRRFs. Complete denitrification is possible with *nirS*-type denitrifiers [36]. Che et al. [39] found a predominance of *nirS*-type over *nirK*-type in eight full-scale municipal WRRFs in different cities of China. Based on regression analysis, Zhang et al. [40] suggested that the abundance of *nirK*-type denitrifiers was correlated with temperature and abundance of *nirS*-type denitrifiers was linearly correlated with both temperature and ammonium concentration.

Both heterotrophic and autotrophic communities of denitrifying bacteria were found, as indicated in Table 3. Heterotrophic denitrifying bacteria (*Ilumatobacter* sp., *Comamonas* sp., *Rhodoferax* sp., *Terrimonas* sp., *Niabella* sp., *Sediminibacterium* sp., *Tistrella* sp., and *Oryzobacter* sp.) are normally found in WRRFs [41,42]. Autotrophic denitrifying bacteria belonging to *Chloroflexi*, *Azospira*, and *Thauera* commonly found in wastewater worldwide were also

present. *Chloroflexi* are the filamentous autotrophic denitrifying bacteria, play a role in sludge flocculation, and are more commonly found in WRRFs designed to remove nutrients, and most appear with a long SRT operation and exposure of the biomass to anaerobic conditions [43]. *Haliscomenobacter* sp. are filamentous bacteria and thrive in phosphorus concentrations [44]. These filamentous bacteria were found and achieved removal of phosphorus in the pre-anoxic process.

Heterotrophic nitrifying bacteria (affiliated to *Pseudomonas* sp.) were found only at H1. Heterotrophic nitrifying bacteria (HNB) have remarkable potential in wastewater BNR engineering fields, and they can also perform aerobic denitrification reactions, directly converting $NH_4^+$ to $N_2$ gas by a single bacterial species [45]. The high ammonium removal efficiency at H1 with the highest COD:TN ratio is likely related to the presence of the HNB.

In this work, DGGE analysis was used for identifying microbial communities responsible for only N removal. However, in further research work, high-throughput sequencing based on 16S rRNA technology or advanced techniques on the next generation sequencing (NGS) with different variable regions would be suggested as the method for microbial analysis instead of DGGE at these full-scale WRRFs.

## 4. Conclusions

With the low COD:N ratio in the influent of L1 and L2 WRRFs, there is insufficient carbon source for denitrifying bacteria. To solve this carbon limitation without external carbon addition, a plant operator has two options. As a first option, the plant operator is able to operate a system with HRT ($\geq$9 h) and long SRT ($\geq$30 d). With a longer SRT, a low DO level (0.9 $\pm$ 0.2 mg-$O_2$/L) is able to be maintained in an aerobic tank. As a second option, the plant operator must operate with high DO level (4.0 $\pm$ 0.5 mg-$O_2$/L), short HRT (1 h), and typical SRT (15–20 d). With a typical SRT, a plant operator needs to keep a significantly low or negligible DO concentration in the anoxic tank for denitrification process to occur.

High N removal performances of full-scale pre-anoxic process at H1 and H2 (high COD:TN ratios of $\geq$8.4) occurred with typical operational parameters: HRT of 9–15 h and SRT of 12–16 d.

A low DO level from 0.7 to 1.7 mg-$O_2$/L at L1 and H1 is responsible for the high abundance of AOA over AOB. *Nitrosospira* could indicate that the long SRT (>30 d) is maintained at L1. In contrast, a high DO (2.4 to 4.5 mg-$O_2$/L) at L2 and H2 contributed to the abundance of AOB over AOA. *Nitrosomonas* were the most abundant and other AOB populations *Nitrosococcus*, *Thiobacillu*, and *Zoogloea* were also present. The WRRF with high COD:TN operated with low DO level facilitated heterotrophic nitrifying bacteria as *Pseudomonas* sp. to high $NH_4^+$ removal efficiency. *Nitrobacter* sp. are more competitive than *Nitrospira* sp. at the low operational DO concentration of L1 and H1. In contrast, the abundance of *Nitrospira* could be higher than the abundance of *Nitrobacter* under the high DO level.

The *nirS* outnumbered *nirK*-type denitrifiers under both the low and high COD:TN conditions. A high abundance of gene-type denitrifiers (*nosZ*) could be found in both WRRFs with low DO concentration. *Chloroflexi*, *Azospira*, *Thauera,* and *Haliscomenobacter* are representative of the autotrophic denitrifying bacterium and *Ilumatobacter*, *Comamonas*, *Rhodoferax*, *Terrimonas*, *Niabella*, *Sediminibacterium*, *Tistrella*, and *Oryzobacter* species would work with the heterotrophic denitrifying bacteria. Maintaining a low DO level during operation of the pre-anoxic process WRRF for saving energy could be possible. However, $N_2O$ gas is able to be produced when maintaining low DO concentration in comparison to when operating at a high DO level. For this reason, future research in $N_2O$ production should be recommended in the full-scale pre-anoxic WRRF at low COD:TN ratios.

**Supplementary Materials:** The following supporting information can be downloaded at: https://www.mdpi.com/article/10.3390/w14050720/s1, Table S1. Efficiency, slope and $r^2$ values of individual real-time PCR assays, Table S2. Overall gene abundance of the with and without pre-anaerobic plants by multiple mean comparisons of one-way ANOVA test.

**Author Contributions:** Performing research, analyzing data, and writing the first draft, S.P.; intiative idea of project including funding acquisition, designing research, troubleshooting, and analyzing data, P.N.; writing—review and editing, P.N., C.-W.L. and K.-H.C. All authors have read and agreed to the published version of the manuscript.

**Funding:** This research work was supported by the NSRF via the Program Management Unit for Human Resources and Institutional Development, Research and Innovation (grant number: B16F630088) and Postdoctoral Fellowship from Kasetsart University Research and Development Institute (KURDI) (received funding support in October 2020 through September 2021).

**Institutional Review Board Statement:** Not applicable.

**Informed Consent Statement:** Not applicable.

**Data Availability Statement:** Not applicable.

**Acknowledgments:** This research work was supported by the NSRF via the Program Management Unit for Human Resources and Institutional Development, Research and Innovation (grant number: B16F630088) and Postdoctoral Fellowship from Kasetsart University Research and Development Institute (KURDI). The authors also would like to thank the Faculty of Engineering, Kasetsart University for their good support and Nimaradee Boonapatcharoen at Pilot Plant Development and Training Institute King Mongkut's University of Technology Thonburi (Bangkuntien) for helpful suggestions on molecular techniques, and Barbara A. Butler, Ph.D. at U.S. Environmental Protection Agency) for final editing.

**Conflicts of Interest:** The authors declare no conflict of interest.

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
