# Peer review of "Efficacies of Nitrogen Removal and Comparisons of Microbial Communities in Full-Scale (Pre-Anoxic Systems) Municipal Water Resource Recovery Facilities at Low and High COD:TN Ratios"

_water, doi:10.3390/w14050720_

Round 1
Reviewer 1 Report
In this study, the authors investigated overall nitrogen removal performance and functional microbial community in four full-scale WWTPs with BNR at high/low C/N ratios. This has much interest for readership especially for designers/operators in wastewater treatment facilties. However, for high-quality research paper, the manuscript seems like a techniquical report, the novelty and new outcomes of this work seems insufficient. Some comments were provided as below. 1. Introduction. Please highlight the significance of your work. Some irrelevant contents should be deleted. 2. In section 2.1. Some key information and the process paramters of full-scale WWTPs in Table 1 should be combined to a new Table for clear demonstration. 3. In 2.3. Nowadays, high-throughput sequencing based on 16S rRNA technology instead of DGGE has been a widely used method for microbial analysis. Unfortunately, it is not done. 4. The correlation between nitrogen removal and influent C/N as well as operating paramters was lacking. This is very important to show the novelty of your reserach. 5. Moreover, the discussion on the difference on function microbial community such as AOB/NOB and denitrifiers among four full-scale WWTPs with different environmental conditions was also needed. 6. In Table 1, operational data should be demonstrated as average±SD. The sampling number also should be indicated. 7. How did C/N ratios affect nitrogen removal and N-removal associated populations? And based on your work, how to provide a guidance for improving nitrogen removal in WWTPs espeically with a low C/N ratio? 8. The new findings from your work were very limited, which restricted the acceptance of your manuscript.Author Response
Review #1
Comments and Suggestions for Authors
In this study, the authors investigated overall nitrogen removal performance and functional microbial community in four full-scale WWTPs with BNR at high/low C/N ratios. This has much interest for readership especially for designers/operators in wastewater treatment facilties. However, for high-quality research paper, the manuscript seems like a techniquical report, the novelty and new outcomes of this work seems insufficient. Some comments were provided as below.
- Introduction. Please highlight the significance of your work. Some irrelevant contents should be deleted.
Ans: We have addressed this comment by rewriting highlight the significance of this work, see detail information below:
When designing an anaerobic system prior to an anoxic and an aerobic system (or w/- and w/o anaerobic at front), wastewater characteristics, such as chemical oxygen demand (COD), total nitrogen (TN), and operational parameters including contact time in the anaerobic tank, the solids retention time (SRT), the hydraulic retention time (HRT), and the DO concentration must be taken into consideration. The proper COD:TN ratio in influent wastewater is an important parameter for biological N removal. In municipal wastewater with a low COD:TN ratio, there is insufficient carbon for the denitrification process resulting in low N removal [2]. External carbon source addition is a significant approach to improve biological N removal (BNR) performance for wastewater with a low COD:TN ratio [3]. However, adding an external carbon source could be expensive in the case of a full-scale WRRF, where there is high capacity. To save costs, operating with longer SRT might be a potential approach to improve biological N removal performance for wastewater with a low COD:TN ratio. Phanwilai et al. [4] achieved significant N removal with a step feed treatment process operated at an SRT >60 d. Liu et al. [5] reported that a system with an SRT at 40 d outperformed systems with shorter SRTs (5, 10, and 20 d).
Maintaining low DO level in the aerobic tank could be another operating parameter to increase BNR performance. In instances with very low DO levels, such as 0–0.5 mg-O2/L, ammonia-oxidizing archaea (AOA) would be the dominant microorganism group responsible for N removal [6]. Increasing the abundance of ammonia-oxidizing bacteria (AOB) was reported with a high DO level of 1.9–3.5 mg-O2/L [6]. The domination of Nitrospira was observed at DO below 1.0 mg-O2/L [7].
Temperature and free ammonia (FA) are also important factors affecting the microbial community. A range of temperature at 10–20ºC was reported to be optimal for Nitrospira [8] and a temperature at 24–25ºC is favorable for Nitrobacter [7]. FA was an inhibitor of nitrite-oxidizing bacteria (NOB) activity [9]. Furthermore, Nitrobacter is more sensitive to FA than Nitrospira [10].
Total nitrogen removal evidence for full-scale (pre-anoxic systems) municipal WRRFs, especially for low and high COD:TN ratios, longer and typical SRTs, and various DO concentrations and temperatures is not available. For this reason, this research focused on a comparison of N removal performance, and identification and quantification of microbial communities from anaerobic, anoxic, and aerobic tanks in four full-scale municipal WRRFs having low (≤5) to high (≥8.4) COD:TN ratios. In this work, only N removal efficiencies with operational parameters (HRT, SRT, and DO level) were discussed by using results from microbial abundance and communities of bacteria related to N removal, such as AOA, AOB, NOB, and denitrifying bacteria (DNB). In addition, the results from this work could be applied to increase N removal efficiencies of other Pre-anoxic w/- and w/o anaerobic WRRFs that have low and high COD:TN ratios in influent.
- In section 2.1. Some key information and the process parameters of full-scale WWTPs in Table 1 should be combined to a new Table for clear demonstration.
Ans: It has been changed and added as suggested by separating in Tables 1 and 2, Table 1 shows operational parameters of the full-scale WWTP and Table 2 shows comparison of the average physical and chemical characteristics of wastewater quality in full-scale Pre-anoxic zone w/- and w/o anaerobic systems by low and high COD:TN ratio (see the revised version)
- In 2.3. Nowadays, high-throughput sequencing based on 16S rRNA technology instead of DGGE has been a widely used method for microbial analysis. Unfortunately, it is not done.
Ans: The reviewer’s understanding is correct. For this reason, the authors mentioned in our further research work, see revised version.
In this work, DGGE analysis was used for identifying microbial communities responsible for only N removal. However, in further research work, high-throughput sequencing based on 16S rRNA technology or advanced techniques on the next generation sequencing (NGS) with different variable regions would be suggested as the method for microbial analysis instead of DGGE at these full-scale WRRFs.
- The correlation between nitrogen removal and influent C/N as well as operating parameters was lacking. This is very important to show the novelty of your research.
Ans: We have addressed this comment as follows.
With the low COD:N ratio in the influent of L1 and L2 WRRFs, there is insufficient carbon source for denitrifying bacteria. To solve this carbon limitation without external carbon addition, a plant operator has two options.
As a first option, the plant operator is able to operate a system with HRT (≥9 h) and long SRT (≥30 d). With a longer SRT, a low DO level (0.9±0.2 mg-O2/L) is able to be maintained in an aerobic tank. The main reason to operate under all these conditions is that partial nitritation and simultaneous nitrification and denitrification (SND) processes are expected to occur in the aerobic tank. Another reason is that, complete biodegradation of organic matter, including readily biodegradable COD (rbCOD) and slow biodegradable COD (sbCOD), and endogenous decay of bacteria could have occurred due to long SRT conditions, which significantly affected the denitrification process.
As a second option, the plant operator has to operate with high DO level (4.0±0.5 mg-O2/L), short HRT (1 h) and typical SRT (15-20 d). With a typical SRT, a plant operator needs to keep a significantly low or negligible DO concentration in the anoxic tank for denitrification process to occur.
- Moreover, the discussion on the difference on function microbial community such as AOB/NOB and denitrifiers among four full-scale WWTPs with different environmental conditions was also needed.
Ans: It has been discussed as suggested (see revised version of manuscript).
The longer SRT and HRT, low DO concentration in the aerobic tank, and high temperature (over 25â—¦C) are the main operating conditions that favor the growth of AOA and AOB communities.
Both the L1 and H1 w/o pre-anaerobic systems had higher AOA abundance, which was expected because the lower DO level, higher temperature, and longer SRT (>30 d) would significantly promote the growth of AOA. This result is similar to the result by Yin et al. [17]. Gao et al. [6] studied the effects of DO levels on the growth of AOB-amoA and AOA-amoA, showing the former is more abundant under high DO levels of 1.9–3.5 mg-O2/L.
- In Table 1, operational data should be demonstrated as average±SD. The sampling number also should be indicated.
Ans: It has been added as suggested (see the revision in Table 2, red color). The total wastewater samples from these four full-scale WRRFs were collected every month (n= 12 samples)
- How did C/N ratios affect nitrogen removal and N-removal associated populations? And based on your work, how to provide a guidance for improving nitrogen removal in WWTPs especially with a low C/N ratio?
Ans: We have addressed the C/N ratios affect nitrogen removal and N-removal associated populations, (see sessions 3.2.1 through 3.2.3). The guidance for improving nitrogen removal in full-scale WRRF with a low C/N ratio, we have addressed this comment, see our response comment (4).
- The new findings from your work were very limited, which restricted the acceptance of your manuscript.
Ans: The authors have addressed this comment, see our response to comment (4).

Reviewer 2 Report
Manuscript ID: water-1561906
Type of manuscript: Article
Title: Efficacies of Nitrogen Removal and Microbial Communities in Full-scale Pre-anoxic Systems Municipal Water Resource Recovery Facilities at Low and High COD:TN Ratios
Authors: Supaporn Phanwilai, Pongsak Lek Noophan *, Chi-Wang Li, Kwang-Ho Choo
This study focused on N removal performance, and identification and quantification of microbes from anaerobic, anoxic, and aerobic tanks in four full-scale municipal WRRFs having low to high COD:TN ratios. The authors concluded that to improved N removal efficiency at a low COD:TN ratio of ≤5 at L1 and L2, a longer SRT (>30 d) and HRT (>8 h) would be recommended. High N removal performances of full-scale Pre-anoxic process at H1 and H2 (high COD:TN ratios of ≥8.4) occurred with typical operational parameters: HRT of 9–15 h, SRT of 12‒16 d. The microbial communities were also investigated. While the results reported could be of interest, the overall quality of the paper is not high enough in my view. There are a lot of grammatical mistakes. In addition, the manuscript is mainly descriptive. There is sometimes a lack of linking sentences. All this contributes to make reading difficult and also sometimes unclear. It is therefore necessary to review the writing thoroughly.
Specific comment:
- It is difficult to understand the title of the article. Pre-anoxic Systems Municipal Water Resource Re-covery Facilities? Efficacies of Nitrogen Removal and Microbial Communities?
- An experimental research abstract, sometimes called a scientific abstract usually includes: The title of the paper. A brief discussion of context or background. The study's objectives--what is the question under discussion? A brief summary of major results and their significance. Main conclusions (or hypothesized conclusions). One sentence discussing the relevance or future directions for research.
- The full name of “AOA, DO” in the Abstract should be added.
- “Temperature and free ammonia (FA) are also important factors …… FA was an inhib-itor of nitrite-oxidizing bacteria (NOB) activity [9]. Furthermore, Nitrobacter is more sen-sitive to FA than Nitrospira [10].” This study focused on N removal performance, and identification and quantifica-tion of microbes from anaerobic, anoxic, and aerobic tanks in four full-scale municipal WRRFs having low (≤5) to high (≥8.4) COD:TN ratios. Hence, the necessity of this paragraph should be reconsidered.
- More discussion should be added in the section 3.1. Major operational parameters and performance of full-scale Pre-anoxic w/o and w/- anaerobic process.
- The influent characteristics of L1 and L2 were different, the performance of the nitrogen removal and microbial communities may be also influenced by the influent characteristics. Some discussion should be added.
- The novelty of the paper should be strengthened.
Author Response
Review #2
Type of manuscript: Article
This study focused on N removal performance, and identification and quantification of microbes from anaerobic, anoxic, and aerobic tanks in four full-scale municipal WRRFs having low to high COD:TN ratios. The authors concluded that to improved N removal efficiency at a low COD:TN ratio of ≤5 at L1 and L2, a longer SRT (>30 d) and HRT (>8 h) would be recommended. High N removal performances of full-scale Pre-anoxic process at H1 and H2 (high COD:TN ratios of ≥8.4) occurred with typical operational parameters: HRT of 9–15 h, SRT of 12‒16 d. The microbial communities were also investigated. While the results reported could be of interest, the overall quality of the paper is not high enough in my view. There are a lot of grammatical mistakes. In addition, the manuscript is mainly descriptive. There is sometimes a lack of linking sentences. All this contributes to make reading difficult and also sometimes unclear. It is therefore necessary to review the writing thoroughly.
Specific comment:
It is difficult to understand the title of the article. Pre-anoxic Systems Municipal Water Resource Recovery Facilities? Efficacies of Nitrogen Removal and Microbial Communities?
Ans: The title has been edited as suggested (see a new title below)
Efficacies of Nitrogen Removal and Comparison of Microbial Communities in Full-scale (Pre-anoxic systems) Municipal Water Resource Recovery Facilities at Low and High COD:TN Ratios
An experimental research abstract, sometimes called a scientific abstract usually includes: The title of the paper. A brief discussion of context or background. The study's objectives--what is the question under discussion? A brief summary of major results and their significance. Main conclusions (or hypothesized conclusions). One sentence discussing the relevance or future directions for research.
Ans: We have addressed this comments (see detail information below).
The novelty of our research is below:
With the low COD:N ratio in the influent of L1 and L2 WRRFs, there is insufficient carbon source for denitrifying bacteria. To solve this carbon limitation without external carbon addition, a plant operator has two options.
As a first option, the plant operator is able to operate a system with HRT (≥9 h) and long SRT (≥30 d). With a longer SRT, a low DO level (0.9±0.2 mg-O2/L) is able to be maintained in an aerobic tank. The main reason to operate under all these conditions is that partial nitritation and simultaneous nitrification and denitrification (SND) processes are expected to occur in the aerobic tank. Another reason is that, complete biodegradation of organic matter, including readily biodegradable COD (rbCOD) and slow biodegradable COD (sbCOD), and endogenous decay of bacteria could have occurred due to long SRT conditions, which significantly affected the denitrification process.
As a second option, the plant operator has to operate with high DO level (4.0±0.5 mg-O2/L), short HRT (1 h) and typical SRT (15-20 d). With a typical SRT, a plant operator needs to keep a significantly low or negligible DO concentration in the anoxic tank for denitrification process to occur.
In future directions for research was suggested in detail below.
In this work, DGGE analysis was used for identifying microbial communities responsible for only N removal. However, in further research work, high-throughput sequencing based on 16S rRNA technology or advanced techniques on the next generation sequencing (NGS) with different variable regions would be suggested as the method for microbial analysis instead of DGGE at these full-scale WRRFs.
The full name of “AOA, DO” in the Abstract should be added.
Ans: The full names have been added to the abbreviations of AOA (Ammonia-oxidizing archaea) and DO (Dissolved oxygen), as suggested.
“Temperature and free ammonia (FA) are also important factors …… FA was an inhib-itor of nitrite-oxidizing bacteria (NOB) activity [9]. Furthermore, Nitrobacter is more sensitive to FA than Nitrospira [10].” This study focused on N removal performance, and identification and quantification of microbes from anaerobic, anoxic, and aerobic tanks in four full-scale municipal WRRFs having low (≤5) to high (≥8.4) COD:TN ratios. Hence, the necessity of this paragraph should be reconsidered.
Ans: It has been reconsidered, as suggested, see detail information below:
Meanwhile, Nitrobacter is more sensitive to free ammonia (FA) concentration compared to Nitrospira [10]. Mehrani et al. [9] reported that FA was a major inhibitor of NOB activity. FA concentrations at L2 (0.17 mg-N/L) and H2 (0.28 mg-N/L) were higher than at L1 (0.15 mg-N/L) and H1 (0.15 mg-N/L). It could be postulated that the FA con-centration was an inhibitor and decreased the abundance of Nitrobacter in these WRRFs w/- anaerobic system, which have lower FA concentrations than L1 and H1.
More discussion should be added in the section 3.1. Major operational parameters and performance of full-scale Pre-anoxic w/o and w/- anaerobic process.
Ans:, This comment was addressed in the manuscript, (see detail information below).
With the low COD:N ratio in the influent of L1 and L2 WRRFs, there is insufficient carbon source for denitrifying bacteria. To solve this carbon limitation without external carbon addition, a plant operator has two options.
As a first option, the plant operator is able to operate a system with HRT (≥9 h) and long SRT (≥30 d). With a longer SRT, a low DO level (0.9±0.2 mg-O2/L) is able to be maintained in an aerobic tank. The main reason to operate under all these conditions is that partial nitritation and simultaneous nitrification and denitrification (SND) processes are expected to occur in the aerobic tank. Another reason is that, complete biodegradation of organic matter, including readily biodegradable COD (rbCOD) and slow biodegradable COD (sbCOD), and endogenous decay of bacteria could have occurred due to long SRT conditions, which significantly affected the denitrification process.
As a second option, the plant operator has to operate with high DO level (4.0±0.5 mg-O2/L), short HRT (1 h) and typical SRT (15-20 d). With a typical SRT, a plant operator needs to keep a significantly low or negligible DO concentration in the anoxic tank for denitrification process to occur.
The influent characteristics of L1 and L2 were different, the performance of the nitrogen removal and microbial communities may be also influenced by the influent characteristics. Some discussion should be added.
Ans: We have addressed this comment (see detail information below).
Longer SRT and HRT, low DO concentration in the aerobic tank, and high temperature (over 25â—¦C) are the main operating conditions favor the growth of AOA community Yin et al. [17].
Both the L1 and H1 w/o pre-anaerobic systems had higher AOA abundance, which was expected because the lower DO level, higher temperature, and longer SRT (>30 d) would significantly promote the growth of AOA. This result is similar to the result by Yin et al. [17]. Gao et al. [6] studied the effects of DO levels on the growth of AOB-amoA and AOA-amoA, showing the former is more abundant under high DO levels of 1.9–3.5 mg-O2/L.
The novelty of the paper should be strengthened.
Ans: We have addressed this comment as follows.
Low COD:N ratio in the influent of L1 and L2 WRRFs, there is insufficient carbon source for denitrifying bacteria. To solve this carbon limitation without external carbon addition, plant operator has two options.
First option, plant operator is able to operate a system with HRT (≥9 h) and long SRT (≥30 d). Associated with longer SRT, low DO level (0.9±0.2 mg-O2/L) is able to maintain in aerobic tank. The main reason to keep under all these conditions is that partial nitritation and simultaneous nitrification and denitrification (SND) processes are expected to occur in aerobic tank. Another possible reason, a complete biodegradation of organic matters including readily biodegradable COD (rbCOD) and slow biodegradable COD (sbCOD) and endogenous decay of bacteria could be occurred due to long SRT condition which significantly affected on denitrification process.
Second option, plant operator has to operate with high DO level (4.0±0.5 mg-O2/L), short HRT (1 h) and typical SRT (15-20 d). Associate with typical SRT, plant operator really needs to keep significantly low DO concentration or negligible in anoxic tank for denitrification process to occur.

Reviewer 3 Report
This study investigated the nitrogen removal performance and microbial community of the full-scale municipal wastewater resource recovery plants which operated in wide ranges of C/N ratio. The experimental results may be useful for other facilities in the near future. However, the research question and the originality of this study were missing. So, it should be said that the paper is not ready for publication.
Introduction
The description is not summarized well. Research questions (e.g. what is known, what is unknown) and/or the necessity of this study should be mentioned.
Results and discussion
The experimental results may be supported the phenomenon occurring in the WRRF. But it was difficult to understand the importance of the result since the originality and motivation of this study were missing as mentioned above. Further, a discussion of how the experimental results give an impact on a research topic (in this case nitrogen removal on WWFR) should be described, which can show the necessity of the study.
Specific comments about the results were below.
-Detail information about the other WRRF (L1, L2, H1) also should be mentioned with a discussion about the effect on WRRF performance.
-It is better to show the MLVSS (MLSS) concentration in each WRRF if it's measured.
-Figure3. Since the MLVSS (or MLSS) concentration was unknown, a comparison between each sample should be carefully conducted to avoid misleading.
Author Response
Reviewer #3
Comments and Suggestions for Authors
This study investigated the nitrogen removal performance and microbial community of the full-scale municipal wastewater resource recovery plants which operated in wide ranges of C/N ratio. The experimental results may be useful for other facilities in the near future. However, the research question and the originality of this study were missing. So, it should be said that the paper is not ready for publication.
Introduction
The description is not summarized well. Research questions (e.g. what is known, what is unknown) and/or the necessity of this study should be mentioned.
Ans: We have addressed this comment by changing and adding more clarification , see detail information below.
When designing an anaerobic system prior to an anoxic and an aerobic system (or w/- and w/o anaerobic at front), wastewater characteristics, such as chemical oxygen demand (COD), total nitrogen (TN), and operational parameters including contact time in the anaerobic tank, the solids retention time (SRT), the hydraulic retention time (HRT), and the DO concentration must be taken into consideration. The proper COD:TN ratio in influent wastewater is an important parameter for biological N removal. In municipal wastewater with a low COD:TN ratio, there is insufficient carbon for the denitrification process resulting in low N removal [2]. External carbon source addition is a significant approach to improve biological N removal (BNR) performance for wastewater with a low COD:TN ratio [3]. However, adding an external carbon source could be expensive in the case of a full-scale WRRF, where there is high capacity. To save costs, operating with longer SRT might be a potential approach to improve biological N removal performance for wastewater with a low COD:TN ratio. Phanwilai et al. [4] achieved significant N removal with a step feed treatment process operated at an SRT >60 d. Liu et al. [5] reported that a system with an SRT at 40 d outperformed systems with shorter SRTs (5, 10, and 20 d).
Maintaining low DO level in the aerobic tank could be another operating parameter to increase BNR performance. In instances with very low DO levels, such as 0–0.5 mg-O2/L, ammonia-oxidizing archaea (AOA) would be the dominant microorganism group responsible for N removal [6]. Increasing the abundance of ammonia-oxidizing bacteria (AOB) was reported with a high DO level of 1.9–3.5 mg-O2/L [6]. The domination of Nitrospira was observed at DO below 1.0 mg-O2/L [7].
Temperature and free ammonia (FA) are also important factors affecting the microbial community. A range of temperature at 10–20ºC was reported to be optimal for Nitrospira [8] and a temperature at 24–25ºC is favorable for Nitrobacter [7]. FA was an inhibitor of nitrite-oxidizing bacteria (NOB) activity [9]. Furthermore, Nitrobacter is more sensitive to FA than Nitrospira [10].
Total nitrogen removal evidence for full-scale (pre-anoxic systems) municipal WRRFs, especially for low and high COD:TN ratios, longer and typical SRTs, and various DO concentrations and temperatures is not available. For this reason, this research focused on a comparison of N removal performance, and identification and quantification of microbial communities from anaerobic, anoxic, and aerobic tanks in four full-scale municipal WRRFs having low (≤5) to high (≥8.4) COD:TN ratios. In this work, only N removal efficiencies with operational parameters (HRT, SRT, and DO level) were discussed by using results from microbial abundance and communities of bacteria related to N removal, such as AOA, AOB, NOB, and denitrifying bacteria (DNB). In addition, the results from this work could be applied to increase N removal efficiencies of other Pre-anoxic w/- and w/o anaerobic WRRFs that have low and high COD:TN ratios in influent.
Results and discussion
The experimental results may be supported the phenomenon occurring in the WRRF. But it was difficult to understand the importance of the result since the originality and motivation of this study were missing as mentioned above. Further, a discussion of how the experimental results give an impact on a research topic (in this case nitrogen removal on WWFR) should be described, which can show the necessity of the study.
Ans: We have addressed this comment, see detailed information below
The novelty of our research is below:
With the low COD:N ratio in the influent of L1 and L2 WRRFs, there is insufficient carbon source for denitrifying bacteria. To solve this carbon limitation without external carbon addition, a plant operator has two options.
As a first option, the plant operator is able to operate a system with HRT (≥9 h) and long SRT (≥30 d). With a longer SRT, a low DO level (0.9±0.2 mg-O2/L) is able to be maintained in an aerobic tank. The main reason to operate under all these conditions is that partial nitritation and simultaneous nitrification and denitrification (SND) processes are expected to occur in the aerobic tank. Another reason is that, complete biodegradation of organic matter, including readily biodegradable COD (rbCOD) and slow biodegradable COD (sbCOD), and endogenous decay of bacteria could have occurred due to long SRT conditions, which significantly affected the denitrification process.
As a second option, the plant operator has to operate with high DO level (4.0±0.5 mg-O2/L), short HRT (1 h) and typical SRT (15-20 d). With a typical SRT, a plant operator needs to keep a significantly low or negligible DO concentration in the anoxic tank for denitrification process to occur.
Specific comments about the results were below.
-Detail information about the other WRRF (L1, L2, H1) also should be mentioned with a discussion about the effect on WRRF performance.
Ans: It has been added as suggested, see detail information below.
The L2 with COD:TN ratio of 4.2 had efficient TN removal of 70%. It is postulated that, plant operator has to operate with high DO level (4.0±0.5 mg-O2/L), short HRT (1 h) and typical SRT (15-20 d). Associate with typical SRT, plant operator really needs to keep significantly low DO concentration or negligible in anoxic tank for denitrification process to occur. In this case, it requires skillful operators to control the system correctly.
To further enhance the removal of NH4+-N, a long SRT of >19 d is recommended because it is assumed that a complete biodegradation of organic matters including readi-ly biodegradable COD (rbCOD) and slow biodegradable COD (sbCOD) and endogenous decay of bacteria could be occurred due to long SRT condition which significantly affected on denitrification process.
To solve this carbon limitation at L1 WRRF without external carbon addition implies that a future operator could operate a system with HRT (≥9 h) and long SRT (≥30 d). Associated with longer SRT, low DO level (0.9±0.2 mg-O2/L) is able to maintain in aerobic tank. The main reason to keep under all these conditions is that partial nitritation and simultaneous nitrification and denitrification (SND) processes are expected to occur in aerobic tank.
-It is better to show the MLVSS (MLSS) concentration in each WRRF if it's measured.
Ans: It has been added in Table 2 as suggested. .
-Figure3. Since the MLVSS (or MLSS) concentration was unknown, a comparison between each sample should be carefully conducted to avoid misleading.
Ans: The reviewer’s understanding is correct. The authors are very careful when we compare microbial abundance of each full-scale municipal WRRF. We always bring other operating parameters and/or results from other researchers to support all comparison statements
For example:
Longer SRT and HRT, low DO concentration in the aerobic tank, and high temperature (over 25â—¦C) are the main operating conditions favor the growth of AOA community Yin et al. [17].
Both the L1 and H1 w/o pre-anaerobic systems had higher AOA abundance, which was expected because the lower DO level, higher temperature, and longer SRT (>30 d) would significantly promote the growth of AOA. This result is similar to the result by Yin et al. [17]. Gao et al. [6] studied the effects of DO levels on the growth of AOB-amoA and AOA-amoA, showing the former is more abundant under high DO levels of 1.9–3.5 mg-O2/L.

Round 2
Reviewer 1 Report
The revision answered the questions of the reviewer.
Author Response
Thank you for your comments.
Reviewer 2 Report
Title: Efficacies of Nitrogen Removal and Microbial Communities in Full-scale
Pre-anoxic Systems Municipal Water Resource Recovery Facilities at Low and
High COD:TN Ratios
Authors: Supaporn Phanwilai, Pongsak Lek Noophan *, Chi-Wang Li, Kwang-Ho Choo
The manuscript was significantly improved. However, some revisions are still necessary before this manuscript can be published. Again, more discussion should be added in the section 3.1. Although the authors have added some discussion about the novelty of this research, the novelty and new outcomes of this work seems still insufficient.
Reviewer 3 Report
The author modified the paper. In my opinion, however, a more logical explanation is needed, especially in the Background. Still, information to explain the motivation and originality is not enough. Details are listed below.
・Meaning of “High COD/N” and “Low COD/N” is hard to understand. There would be a suitable COD/TN ratio for A2O or AO systems. So, the author must be explained the suitable condition to occur BNR in such a system with showing the exact value (not as in “high” or “low”). Then, reader can understand what is “High” and “Low”.
・Similarly, it is hard to understand how the COD/TN ratio of the selected plant differs from the general setting (recommended setting). If the COD/TN ratio in the selected plant was out of range from the general setting, this is unique and would be a research motivation for evaluating its performance.
Figure 3.
The author had MLVSS information. So, the functional genes should be calculated in copies/MLVSS, not in ml-sludge. Then, the comparison between each plant and genes can be conducted. If not, the comparison between each plant should be carefully conducted and explained.
Round 3
Reviewer 2 Report
As the authors has made the revision, I think the manuscript can be accepted now.
Reviewer 3 Report
The author revised the manuscript. So, the paper can be accepted.